# The Motion of the Italian National Bioethics Committee on Aggressive Treatment towards Children with Limited Life Expectancy

**DOI:** 10.3390/healthcare8040448

**Published:** 2020-11-01

**Authors:** Matteo Bolcato, Marianna Russo, Alessandro Feola, Bruno Della Pietra, Camilla Tettamanti, Alessandro Bonsignore, Rosagemma Ciliberti, Daniele Rodriguez, Anna Aprile

**Affiliations:** 1Department of Molecular Medicine, University of Padua, 35121 Padua, Italy; mariannarusso.medleg@gmail.com (M.R.); danielec.rodriguez@gmail.com (D.R.); anna.aprile@unipd.it (A.A.); 2Department of Experimental Medicine, University of Campania “Luigi Vanvitelli”, 80138 Naples, Italy; alessandro.feola@unicampania.it (A.F.); bruno.dellapietra@unicampania.it (B.D.P.); 3Department of Health Sciences, Section of Legal and Forensic Medicine, University of Genova, 16126 Genova, Italy; camilla.tettamanti85@gmail.com (C.T.); alessandro.bonsignore@unige.it (A.B.); 4Department of Health Sciences, Section of History of Medicine and Bioethics, University of Genova, 16126 Genova, Italy; rosellaciliberti@yahoo.it

**Keywords:** Italian National Bioethics Committee, young children, short life expectancy, aggressive treatments, therapeutic obstinacy

## Abstract

The motion of the Italian National Bioethics Committee entitled “Aggressive treatment or therapeutic obstinacy on young children with limited life expectancy” comprises a premise that rejects therapeutic obstinacy and makes 12 recommendations. Recommendation no. 1 states the general rules: it ascribes a cardinal role to a shared care plan, it supports pain management therapy and pain relief, it opposes ineffective and disproportionate clinical treatment and defensive medicine. The other recommendations are correlated to the enacting of a national law establishing clinical ethics committees in paediatric hospitals; participation of parents and their fiduciaries in the decision-making processes; recourse to courts only as extrema ratio in the event of irremediable disagreement between the medical team and the family members; accompaniment at the end of life also through continuous deep sedation combined with pain therapy; access to palliative care; the need to reinforce research on pain and suffering in children; clinical trials and research studies conducted in children; the training of doctors, healthcare personnel and psychologists, to support parents in emotional and practical terms; the facilitation of the closeness of parents to children in extremely precarious clinical conditions; the relevant role of the associations of parents of sick children. Comments are made, in particular, about the innovative recommendations respectively relating to the adoption of care planning, the establishment, by law, of clinical ethics committees in paediatric hospitals and the limitation of recourse to courts—only as *extrema ratio*—in the event of irremediable disagreement between the medical team and the family members.

## 1. Introduction

The Italian National Bioethics Committee (CNB), established by decree of the president of the council of ministers of 28 March 1990, is an advisory body of the government, parliament and other institutions that formulates opinions and points out solutions also with a view to drawing up legislative acts on ethical issues arising from the advancement of research studies and with the new technological applications in the context of life sciences and healthcare. The CNB also carries out activity at a supranational level with regular meetings with European ethics committees and ethics committees worldwide. The committee expresses its indications through opinions, motions and replies. Opinions and replies are highly complex documents that provide for complex procedures. An initial text drawn up as a draft during the discussion within a dedicated working group is subsequently brought to the attention of the plenary session, that amends its content, discussing and approving its final version. In general, motions are more concise documents that, given their pressing nature, are approved with greater swiftness [1].

On 30 January 2020, the CNB issued a motion entitled “Aggressive treatment or therapeutic obstinacy of treatment on young children with limited life expectancy”. After an analytical premise, it sets out 12 recommendations concerning the care of these children and the relationship to be established with the parents in order to provide the necessary information and to support them in the evaluation process and final decision. The 12 recommendations are made available on the Italian government’s website [2] and are listed in English in Box 1.

Box 1The recommendations contained in the motion of the Italian National Bioethics Committee dated 30 January 2020, entitled “Aggressive treatment or therapeutic obstinacy on young children with limited life expectancy”.**1.** Identify aggressive treatment through scientific and clinical data that are as objective as possible, ensuring the best quality of available treatment and possibly relying on shared planning between the medical team and the parents, in the best interests of the child. The best interests of the child is the guiding criterion in this situation and must be defined starting with the temporary clinical condition, coupled with the consideration of pain and suffering (insofar as it is possible to measure them), and of respect of his/her dignity, excluding any evaluation in terms of economic costs. Doctors should avoid going down ineffective and disproportionate clinical pathways just to consent to the requests of parents and/or to fulfil criteria of defensive medicine.**2.** Establish and make effectively operational the clinical ethics committees in paediatric hospitals by national law, with an advisory and educational role, so as to favour the assessment of the complexity of such decisions and to search for mediation in controversies arising between doctors and parents. These ethics committees shall be interdisciplinary and composed of paediatricians, medical experts in the area of expertise under scrutiny, nurses and non-healthcare professionals such as bioethics experts and biojurists. The committees shall not replace the professionals in taking the appropriate decisions, but rather help them exercise their autonomy in a responsible manner.**3.** Integrate the decision-making processes of doctors and ethics committees with the participation of parents and their fiduciaries, ensuring space and time of communication and reflection, involving them in the care and treatment plan of the young patient, adequately informing them about the possible progression of the ongoing pathology in order to pinpoint the limits of therapeutic intervention and the legitimacy of commencing, maintaining or discontinuing treatment by resorting to palliative care.**4.** Allow for a potential second opinion, in addition to the one expressed by the team first admitting the patient into care, if requested by the parents or by the treating team, ensuring the freedom of choice of the parents, under conditions of scientific authority, taking into consideration the best interests of the child. The CNB hopes that two opinions might provide greater certainty of the identification of aggressive treatment and greater sharing in the commencement, continuation, or discontinuation of ongoing treatment. To this end, the entire clinical documentation of the patient (medical record and electronic health record) should be made easily accessible so as to evaluate the scientific and ethical opinion concerning the alleged aggressive treatment.**5.** Provides for recourse to the court, in the event of irremediable disagreement between the medical team and the family members, as *extrema ratio*, pursuant to Law no. 219/2017; this solution should only be considered after seeking mediation through an adequate communication with the parents or the family, taking into consideration a correct clinical documentation and the request to the clinical ethics committee.**6.** Prevent the prohibited unreasonable obstinacy of treatment from resulting in the abandonment of the child, towards whom the doctors are entirely bound by an obligation of provision of appropriate treatment and supports, whether technological or pharmacological, and of palliative care with accompaniment at end of life, also through continuous deep sedation combined with pain therapy.**7.** Ensure access to palliative care in hospital and at home alike, homogeneously across the territory.**8.** Increase research on pain and suffering in children, in order to implement and improve the validation of objective rating scales measuring pain and suffering, able to orient clinical decisions, combined with other parameters.**9.** Prevent the child, especially with an inauspicious short-term prognosis, from being considered as the mere object of clinical trials and research studies by the doctors.**10.** Implement the training of doctors, healthcare personnel and psychologists, to create a substantial core of professionals able to support the parents in emotional and practical terms alike (social workers, psychologists, bioethics experts, family associations, voluntary associations) and accompany them on their difficult journey brought about by the illness of their child.**11.** Facilitate the closeness of the parents to children under extremely precarious clinical conditions (e.g., palliative care administered at home, expected sick leave, and so on).**12.** Acknowledge the significant role played by the associations of parents of sick children and to reinforce networks for solidarity towards parents also on the part of society.

## 2. Global Content and Recipients of the “Recommendations”

The premise of the “Motion”, that is entirely dedicated to “aggressive treatment” [3,4,5], shows that the document is aimed at opposing this practice, at times implemented “almost instinctively” since best efforts are made to ensure that everything possible is done, also at the request of parents, and without considering the negative effects that this might have on the child in terms of suffering [6,7,8]; at times implemented by some professionals “not to ensure the health and well-being of the patient, but as a form of protection and guarantee of their responsibilities” without realising that this assumption is entirely unsubstantiated since the interests of the doctor cannot come before those of the child [9,10,11,12].

By “aggressive treatment” the CNB means “the commencement of treatment which is presumed to be ineffective or the continuation of treatment that has become of documented ineffectiveness in relation to the goals of care of the sick person or of improvement in his/her quality of life (intended as well-being) or such as to cause further suffering to the patient and a precarious and distressing life prolongation without any further benefits”.

In light of the foregoing, the “Motion” would appear to be directed at doctors, but instead it broadens the scope of recipients by explicitly also including other healthcare professionals, as well as institutional subjects, the latter not being specified but easily identifiable based on the content of the single recommendations that entail the creation and coordination of specific services and activities via encoded formal rules (laws and regulations) pertaining to the state, the regions or the healthcare authorities on a case-by-case basis. This results in a set of integrated indications that ensure, on the one hand, the necessary organisational-managerial support and, on the other hand, competent operation on the part of the professionals.

Recommendation 1 presents the key rules in narrative form:The benchmarks, that can be identified as follows: scientific and clinical data that are as objective as possible, the best interests of the child [13], consideration of his/her pain and suffering and respect for his/her dignity [14];Behaviour to be rejected: aggressive treatment (and the correlated ineffective and disproportionate clinical pathways), requests from parents in this sense, defensive medicine, assessments in terms of economic costs [15,16,17].

The subsequent recommendations stem from these general rules and are more specifically relative to the implementation or performance of certain activities by, as mentioned previously, diversified subjects, namely institutions and professionals.

The recommendations aimed at the aforementioned institutions concern: the creation of clinical ethics committees in paediatric hospitals (recommendation 2), the guarantee to be able to access palliative care [18,19] (recommendation 7), increased pain research [20,21] (recommendation 8), the dignity of the child in clinical trials (recommendation 9), the implementation of training of healthcare professionals in general (recommendation 10), facilitation of the closeness of parents to their children (recommendation 11), the acknowledgement of the role of the associations of parents of sick children (recommendation 12).

These recommendations are directed at institutions, however, the commitment of healthcare professionals in the promotion of or participation in initiatives fostering and/or stimulating institutional intervention and/or the application of potential regulations after their approval is implicit. For instance, recommendation 10, referring to the implementation of training, should also be interpreted in the sense that healthcare professionals must make an effort to request such training.

Again, for example, an organisation that facilitates access to palliative care, in accordance with recommendation 7, is pointless if doctors do not actually promote such care.

Furthermore, increased research on pain in children, as per recommendation 8, means making available funds and human resources to conduct this research, but also that individual professionals must be able to take appropriate initiatives so as to contribute, with their experience, their activity and working hypotheses, to the design and development of this research.

Recommendations aimed at healthcare professionals, and some in particular at doctors, include: encouraging reliance on a plan shared [22,23] with the parents in the best interests of the child (recommendation 1), integrating the decision-making processes of doctors with the participation of parents and their fiduciaries (recommendation 3), reference to the promotion of a potential second opinion (recommendation 4), the limitation of recourse to the courts as extrema ratio, in the event of irremediable disagreement between the medical team and the family members (recommendation 5), the absolute duty to provide appropriate treatment and support and palliative care with accompaniment at the end of life, also through continuous deep sedation combined with pain therapy (recommendation 6).

Above all, some of the recommendations targeting healthcare professionals are to be viewed, in some respects, as also aimed at the pertinent institutions. The systematic organisation of recourse to shared planning, with reference to recommendations 1 and 3, is out of the question if the conditions are not provided for “ensuring space and time for the communication of reflection” [24]. Even the activation of a second opinion, under the responsibility of the doctor in full respect of the parents’ wishes, is only feasible if healthcare facilities draw up rules that allow there to be useful tools, for instance, starting from the possibility of making the clinical documentation mentioned in recommendation 4 available to the doctor requested to as a consultant. Recommendation 12 is to be considered as directed, without distinction, at all potential recipients: institutions and doctors alike are indeed called upon to recognise “the significant role played by the associations of parents of sick children and to reinforce networks for solidarity towards parents also on the part of society”.

## 3. Innovative Content in the “Recommendations”

With regard to some of the valuable content of the “Recommendations”, three innovative reflections are proposed: shared planning (recommendation 1), clinical ethics committees in paediatric hospitals (recommendation 2) and recourse to courts only as extrema ratio (recommendation 5).

### 3.1. Shared Planning

Shared care planning was introduced in Italy with art. 5 of Law no. 219 of 22 December 2017 [25,26,27,28,29], and pursuant to paragraph 9 of art. 1 of the same law its implementation is guaranteed in every healthcare facility. In accordance with recommendation 1, shared planning aims to select the most appropriate ways of taking care of a sick child. In pursuance of the aforementioned art. 5, recommendation 1 provides that the entire “medical team” should take part in the shared planning. This planning must be promoted in the best interests of minors and the sole “possibility” of not implementing it is given by the reasoned refusal of the parents who, in any case, must be suitably informed about this procedure. It is accomplished “with the participation of parents and their fiduciaries, ensuring space and time of communication and reflection, involving them in the care and treatment plan of the young patient, adequately informing them about the possible progression of the ongoing pathology in order to pinpoint the limits of therapeutic intervention and the legitimacy of commencing, maintaining or discontinuing treatment by recurring to palliative care” (recommendation 3, to be deemed as relevant to recommendation 1).

Shared care planning is the appropriate procedure for a constructive discussion between medical team and parents.

In this context, the best interests of the child must be addressed, taking into consideration the pathology characteristics, the potential and the limits of the proposed treatment, the expected benefits and the costs, understood in the broad sense. Each medical action to be planned must fulfil criteria of appropriateness and proportionality; interventions solely aimed at prolonging physical life that fail to take into account the interests of the patient are not respectful of personal dignity. In adults, the proportionality between costs and benefits also considers the personal judgement of the subject, who interprets and measures it in the light of his/her own values and own vision of life. In the case of a child, these aspects are usually assessed by his/her parents, as his/her natural guardians, naturally taking into account his/her needs, aspirations and interests.

In the case of children with limited life expectancy, for whom ineffective or inappropriate, and often onerous, treatment is envisaged, the parents’ assessment becomes particularly complex and it is crucial that it should be based on analytical clinical elements and on a defined care plan, clearly outlined by the treating team. The latter shall also accompany the parents adequately informing them about the possible progression of the ongoing pathology in order to pinpoint the limits of therapeutic interventions and the legitimacy of commencing, maintaining, or discontinuing treatments, including therapies with both life-prolonging and palliative intent.

The discussion between the treating team and the parents could give rise to critical aspects also correlated to some of the fundamental concepts to which reference should be made to orient choices. For instance, the very notion of “best interests”, referred to in recommendation 1, is not easy to define or grasp, especially in the case of a very young child. The diverse elements characterising the interests can be enhanced by the medical team in a different way as compared to the parents. It is possible that doctors tend to rely on biological parameters (integrity, functionality, suffering, pain, conscience) whereas parents rely on other anthropological-emotional-relational parameters (being the object and subject of affection also for more people aside from the parents, relationships with others, desires, experiences, self-expression, learning, change management and so forth).

Doctors and parents taking part in the planning of treatment may, therefore, not have the same idea about the child’s interests. Accordingly, parents might develop an opinion in favour of intensive/invasive treatments presuming to know the interests of the child, based on anthropological-emotional-relational criteria, conversely denied by the doctors on the basis of biological criteria. In the event of discrepancies in the assessment of best interests, it should be considered that only the parent perceive the emotional and relational elements, according to their relationship of shared life experience with the child and, thus, their assessment is to be considered as predominant, provided it is expressed following in-depth meetings aimed at allowing them to glean the impact on the child of the clinical activities and biological implications leading the doctors to a different evaluation of the best interests.

If, for the purposes of the choice, the efficacy of a specific treatment should be evaluated, the opinion, necessarily rooted in science, shall be purely medical; the inefficacy of treatment should be conveniently illustrated to the parents, always bearing in mind their difficult situation and possibly proposing to resort to a second opinion, as provided for by recommendation 4. Ineffective treatment, despite being requested by the parents, cannot be accepted in the planning of treatment.

The systematic adoption of a shared care plan will also be endowed an educational value for those doctors who are inclined, for the purposes of “defensive medicine”, to passively welcome any requests (even unreasonable) advanced by the parents. It is likely that this tool could prove useful also to oppose potential legal action initiated by the parents, since adequate and shared verbalisation of the care plan can prove the inconsistency of any potential grievances.

### 3.2. Clinical Ethics Committees in Paediatric Hospitals

The clinical ethics committees existing in Italy were established through local initiatives and are not contemplated by state laws. This also applies to paediatric ethics committees. Hence, the exhortation to “establish, by national law,” clinical ethics committees in paediatric hospitals is especially innovative. Recommendation 2 delineates the characteristics of the committees, which should, correctly, be limited to performing purely advisory and educational roles, both for the doctors (but also for the other healthcare professionals) and parents alike. The activity of the committee refers to “the assessment of the complexity” of the decisions and the search for “a mediation in controversies arising between doctors and parents”. The “mediation” activity of the ethics committee is questionable, since intervention aimed at settling a potential dispute is in contrast with the advisory role previously correctly declared. Reference to the fact that the aforementioned ethics committees are interdisciplinary in nature is apt, even though the list of its members, as suggested in recommendation 2, is insufficient: for instance, there is a lack of, inter alia, certain healthcare professionals often involved in the care of the terminal child such as physiotherapists and dieticians, as well as representatives of citizens, medico-legal experts, psychologists and social workers. The proposed list of members is, therefore, to be viewed merely as illustrative in nature.

### 3.3. Recourse to Courts Applied only as *Extrema Ratio*

Recourse to the Courts by some doctors, in the case of their disagreement with family members, is frequent in Italy and is substantiated by the entirely inaccurate hypothesis that the court called upon may have the jurisdiction to solve a problem that is partly scientific and partly ethical in nature, entailing complex decisions or, even worse, by the even more perverse hypothesis that the doctor may, in any case, delegate to the court responsibility for the decision. Recommendation 5 intends to reduce recourse to courts by the doctors, limiting this to cases of “irremediable” disagreement between the medical team and the family members and “after seeking mediation through an adequate communication with the parents or the family, taking into consideration a correct clinical documentation and the request to the clinical ethics committee.” The Latin phrase, *extrema ratio*, should be viewed as the last resort of the doctor, to be carried out only and exclusively following the irreversible failure of all the activities to be conducted in the relationship of care in its complexity, and not outside the latter, in the hopes that the court may assume the role of *deus ex machina*. Activities to be carried out before resorting to the court include the request to perform a second opinion as provided for by recommendation 4.

Recommendation 5 provides that recourse to the Court should, in any case, take place, pursuant to Law no. 219/2017; this law considers the subject of art. 3, paragraph 4, that, however, only considers the case whereby “the legal representative of the minor rejects the proposed treatments whereas the doctor believes these to be appropriate and necessary”. The conflict to which recommendation 5 refers lies in the irremediable disagreement between medical team and parents stemming from their request to continue or commence treatment that is deemed inappropriate by the doctor.

An alternative to resorting to the tutelary judge for the doctor may be the choice to refrain from administering unreasonable treatment, even if requested by the parents, specifically detailing the reasons behind it in the healthcare documentation [30,31,32]. Recourse to the court appears to be especially superfluous if the “irremediable conflict” is related to the efficacy of a given treatment, denied, as substantiated, by the treating team and by those expressing a second opinion. The provision of paragraph 6 art. 1 of Law no. 219/2017 supports this option of not resorting to the court, whereby “The patient cannot demand healthcare treatment in breach of the law, of professional ethics or good clinical practices: the doctor has no professional obligations towards such requests”. Likewise, the patient cannot demand for him/herself treatments in breach of a good professional conduct, as is also the case for the legal representative of the minor who is the patient.

The *extrema ratio* of recourse to the court should therefore be correlated to irremediable conflicts concerning appropriateness of treatment and assessment of the best interests of the child.

## 4. Conclusions

The widespread dissemination in Italy of this motion of the CNB among healthcare professionals and persons responsible for healthcare policies is desirable, as it contains recommendations useful as guidance to favour and orient their thoughts on the efficacy, appropriateness and proportionality criteria to be adopted, in their respective roles, in full respect of the dignity of children with limited life expectancy.

However, with regard to doctors, knowledge of the “Motion” is not sufficient as, in relation to a young sick child, it is imperative to be respectful of his/her dignity and to be able to identify his/her best interests considering, on the one hand, the pathology characteristics, the potential and the limits of the possible treatments but also, on the other hand, the anthropological-emotional-relational parameters that only parents are able to perceive and assess, and also comparing them to the negative consequences of treatment.

It is no coincidence that parents, as bearers of the perception of the interests of the young child, are valued in the “Motion” advanced by the CNB.

Recommendation 12 completes the “Motion”, by suggesting that the associations of parents of sick children be granted a “relevant role” and that “networks for solidarity towards parents also on the part of society” be reinforced.

The assessment of parents as compared to the choices to be made in the interests of children with limited life expectancy are particularly complex not only intrinsically, but also because the care plan of the young child has repercussions on the relationship of the parents and daily management of the household which can include the sisters and brothers of the sick child as well as on the global life project.

Hence, recommendation 12 aptly provides for a commitment on the part of society in support of the needs of parents burdened by such choices. Recommendation 10 should be interpreted in this sense in that it contemplates that the required training not only of doctors but also of professionals such as social workers, psychologists and bioethics experts, should equip them with the skills to support parents in emotional and practical terms and accompany them on their difficult journey.

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
