# Peer review of "The Motion of the Italian National Bioethics Committee on Aggressive Treatment towards Children with Limited Life Expectancy"

_healthcare, 2020, doi:10.3390/healthcare8040448_

Round 1

Reviewer 1 Report

The article is interesting from an ethical point of view and also has a clinical and application point of view for the doctor. The topic is dealt with rigorously and in a methodologically correct manner. All the points included in the document of the Bioethics Committee are subject to in-depth analysis and criticism. Only one aspect needs to be integrated, namely the citations of the literature on the same topic. In fact, there is a work that concerns the same topic and which should be cited in line 119, after citation number 11, and is the following: Fedeli et al, Open Medicine, 2020, 15; 515-519, The will fa young minor in terminal stage of sikness: a case report. With this supplement, minor revision, the work can be published.

Author Response

The article is interesting from an ethical point of view and also has a clinical and application point of view for the doctor. The topic is dealt with rigorously and in a methodologically correct manner. All the points included in the document of the Bioethics Committee are subject to in-depth analysis and criticism. Only one aspect needs to be integrated, namely the citations of the literature on the same topic. In fact, there is a work that concerns the same topic and which should be cited in line 119, after citation number 11, and is the following: Fedeli et al, Open Medicine, 2020, 15; 515-519, The will fa young minor in terminal stage of sikness: a case report. With this supplement, minor revision, the work can be published.

We thank the reviewer for his helpful comments which allowed us to improve the manuscript. As indicated we have added the follow references in line 119, which is a useful element in the article.

  • Fedeli, P., Giorgetti, S., Cannovo, N. (2020). The will of young minors in the terminal stage of sickness: A case report, Open Medicine15(1), 513-519. doi: https://doi.org/10.1515/med-2020-0152

Reviewer 2 Report

Thank you for your work.  These recommendations are excellent and much can be accomplished by their broad adoption.  Your review highlights and adds useful insight into these recommendations.

Thoughts:

1) I wonder who is the intended audience for this paper?

I think if the intended audience is clinical ethicists, policy-makers, and medico-legal professionals, this language is likely very familiar, clear, and straightforward. 

However, I can say with confidence that most direct health care providers that I know would not be able to follow some of the discussion. Example:

"In the event of discrepancies in the assessment of best interests, it should be considered that only the parent perceive the anthropological-emotional-relational elements, according to their relationship of shared life experience with the child and, thus, their assessment is to be considered as predominant, provided it is expressed following in-depth meetings aimed at allowing them to glean the impact on the child of the biological parameters leading the doctors to a different evaluation of the best interests."

There is absolutely nothing wrong with this sentence. It is just simply not how direct care providers communicate, either in person or in academic papers. I do fear that many direct care providers will stop reading when they see language that resembles legal contracts (e.g. "... art 5 of Law no. 2019 of 22 December 2017, and pursuant to paragraph 9 of art. 1 of the same law its implementation...")

2) The only specific concern regarding messaging is perhaps too late. Recommendation 3 refers to "resorting to palliative care". Perhaps this just does not translate well from Italian to English.  But in English, it poorly captures the role of palliative care. Palliative care throughout the course of a patient's life-threatening condition and is often in parallel with life-prolonging therapies.  I would have considered a statement such as: 

"... adequately informing them about the possible progression of the ongoing pathology in order to pinpoint the limits of therapeutic interventions and the legitimacy of commencing, maintaining, or discontinuing treatments, including therapies with both life-prolonging and palliative intent."

Otherwise, I think these recommendations are excellent. There is interesting discussion about their novelty and implementation. As written, however, I think it would be of interest an important, but select audience.  

Author Response

Thank you for your work.  These recommendations are excellent and much can be accomplished by their broad adoption.  Your review highlights and adds useful insight into these recommendations.

Thoughts:

1) I wonder who is the intended audience for this paper?

I think if the intended audience is clinical ethicists, policy-makers, and medico-legal professionals, this language is likely very familiar, clear, and straightforward.

However, I can say with confidence that most direct health care providers that I know would not be able to follow some of the discussion. Example:

"In the event of discrepancies in the assessment of best interests, it should be considered that only the parent perceive the emotional and relational elements, according to their relationship of shared life experience with the child and, thus, their assessment is to be considered as predominant, provided it is expressed following in-depth meetings aimed at allowing them to glean the impact on the child of the clinical activities and biological implications leading the doctors to a different evaluation of the best interests."

There is absolutely nothing wrong with this sentence. It is just simply not how direct care providers communicate, either in person or in academic papers. I do fear that many direct care providers will stop reading when they see language that resembles legal contracts (e.g. "... art 5 of Law no. 2019 of 22 December 2017, and pursuant to paragraph 9 of art. 1 of the same law its implementation...")

We want to thank the reviewer for his efforts and comments that have allowed us to improve the manuscript.

Although the article is aimed at personnel in the sector, the adoption of a more understandable and direct language can certainly help in the use of the recommendations. For this reason we have modified in a more understandable way to lines 233-238. However, we have kept the references to law 219 as we think they can be important to make known a recent legislative intervention that is still little known and used but important in a national and European context.

2) The only specific concern regarding messaging is perhaps too late. Recommendation 3 refers to "resorting to palliative care". Perhaps this just does not translate well from Italian to English.  But in English, it poorly captures the role of palliative care. Palliative care throughout the course of a patient's life-threatening condition and is often in parallel with life-prolonging therapies.  I would have considered a statement such as:

"... adequately informing them about the possible progression of the ongoing pathology in order to pinpoint the limits of therapeutic interventions and the legitimacy of commencing, maintaining, or discontinuing treatments, including therapies with both life-prolonging and palliative intent."

Otherwise, I think these recommendations are excellent. There is interesting discussion about their novelty and implementation. As written, however, I think it would be of interest an important, but select audience.

 We want to thank the reviewer again for his efforts and comments that have allowed us to improve the manuscript.

We agree with what he indicated about this specific topic, we wanted to insert the suggested sentence (line 2015-2018) to better explain the point despite the fact that the recommendations cannot be changed.

Thanks for your interest

This manuscript is a resubmission of an earlier submission. The following is a list of the peer review reports and author responses from that submission.